# Preparation and Characterization of Photoactive Anatase TiO$_2$ from Algae Bloomed Surface Water

**Sayed Mukit Hossain [1]**, **Heeju Park [2]**, **Hui-Ju Kang [3]**, **Jong Beom Kim [2]**, **Leonard Tijing [1]**, **Inkyu Rhee [4]**, **Young-Si Jun [3]**, **Ho Kyong Shon [1],\*** and **Jong-Ho Kim [2],\***

[1] Faculty of Engineering and IT, University of Technology, Sydney, P.O. Box 123, Broadway, NSW 2007, Australia; sayed.m.hossain@student.uts.edu.au (S.M.H.); leonard.tijing@uts.edu.au (L.T.)

[2] School of Chemical Engineering, Chonnam National University, Gwangju 61186, Korea; point1014@hanmail.net (H.P.); mask-k@hanmail.net (J.B.K.)

[3] Department of Advanced Chemicals & Engineering, Chonnam National University, 77 Yongbong-ro, Buck-gu, Gwangju 61186, Korea; gmlwn120@gmail.com (H.-J.K.); youngsi.jun@gmail.com (Y.-S.J.)

[4] Department of Civil Engineering, Chonnam National University, Gwangju 61186, Korea; rheei@chonnam.ac.kr

\* Correspondence: Hokyong.Shon-1@uts.edu.au (H.K.S.); jonghkim@chonnam.ac.kr (J.-H.K.); Tel.: +61-447-332-707 (H.K.S.)

**Abstract:** The purpose of the study was to effectively treat algae bloomed water while using a Ti-based coagulant (TiCl$_4$) and recover photoactive novel anatase TiO$_2$ from the flocculated sludge. Conventional jar tests were conducted in order to evaluate the coagulation efficiency, and TiCl$_4$ was found superior compared to commercially available poly aluminum chloride (PAC). At a dose of 0.3 g Ti/L, the removal rate of turbidity, chemical oxygen demand (COD), and total phosphorus (TP) were measured as 99.8%, 66.7%, and 96.9%, respectively. Besides, TiO$_2$ nanoparticles (NPs) were recovered from the flocculated sludge and scanning electron microscope (SEM), energy dispersive X-ray spectroscope (EDX), and X-ray diffraction (XRD) analysis confirmed the presence of only anatase phase. The recovered TiO$_2$ was found to be effective in removing gaseous CH$_3$CHO and NO$_x$ under UV-A lamp at a light intensity of 10 W/m$^2$. Additionally, the TiO$_2$ mixed mortar blocks that were prepared in this study successfully removed atmospheric nitrogen oxide (NO$_x$) under UV irradiance. This study is one of the first to prepare anatase TiO$_2$ from flocculated algal sludge and it showed promising results. Further research on this novel TiO$_2$ concerning internal chemical bonds and shift in the absorbance spectrum could explore several practical implications.

**Keywords:** resource recovery; NO$_x$ removal; coagulation/flocculation; photocatalyst; titania synthesis

## 1. Introduction

Surface water is one of the primary sources of natural water resources for fulfilling a prominent portion of diverse water demands around the world. However, a major concern for surface water is rapid algal bloom that results from eutrophication, due to seasonal variation in the physicochemical properties of surface water along with numerous anthropogenic activities (e.g., uncontrolled dredging, inappropriate effluent discharge, etc.) [1,2]. Consequently, algae enriched water could facilitate the generation of various algal organic matter (AOM), which might include both extracellular organic matters (EOM) and intercellular organic matters (IOM) [3]. Algae cells, together with corresponding EOMs and IOMs, can cause the development of disinfection by-products (DBPs), toxins (e.g., hepatoxins and neurotoxins), and unpleasant tastes and odors in surface water, resulting in the significant deterioration of water quality [4,5]. In general, negatively charged algae forms a heterogeneous stable

colloidal suspension in eutrophic water [6]. Besides, the low structural rigidity and high hydrophilicity of algal cells make them very difficult to remove from water [3,7]. Consequently, conventional treatment facilities face plenty of convoluted difficulties, while treating algae-induced water, (e.g., requirement of higher coagulant dose, clogging of filter media, the formation of DBPs from cell lysis, and microbial regrowth in the effluent side of the distribution system) [8,9].

To date, a number of treatment processes have been developed to alleviate or remove algal cells from surface water. Broadly, the prevailing methods can be categorized as: (a) physical processes, such as dissolved air floatation (DAF), direct media filtration and membrane filtration [10]; (b) chemical process (both inorganic and organic), such as coagulation/flocculation and pre-oxidation (using chlorine, ozone, permanganate), EOM as flocculation aid [11]; and lastly, (c) electromagnetic irradiation, such as UV irradiated disinfection [12] and ultrasonic irradiation assisted algae removal [13]. However, from the aspect of economic limitations and practical implications, chemical processes have been used conventionally to treat algae enriched water. More precisely, coagulation/flocculation is the mostly used method for the removal of algal cells [2,6]. Nevertheless, the conventional coagulation/flocculation (using Al and Fe based coagulants) produces a voluminous amount of sludge in addition to residual impacts of the trace metals (Al and Fe), needing extensive post-treatment and appropriate disposal schemes [14]. In other words, the corresponding treatment train gets convoluted. Later, Ti-based coagulants have been used as an alternative to conventional Al and Fe based coagulants to overcome this difficulty, and previous studies reported superior coagulation efficiencies that were related to physicochemical properties of effluent and floc morphology [15,16].

Chekli et al. [17] reported a comparable coagulation performance of Ti-based coagulants in algal turbid water compared to $FeCl_3$ and argued that at a lower coagulant dose (<9 mg/L), $TiCl_4$ performed better to remove turbidity. Moreover, a related study [18] affirmed the superior coagulation efficiency of $TiCl_4$ over $FeCl_3$ in algae augmented synthetic seawater and reported more than 90% removal of EOM, which generated low molecular weight organics (LMWs). Recently, Xu et al. [14] used $TiCl_4$ to conduct flocculation in laboratory simulated algae ($10^6$ cells/mL) contained reservoir water and demonstrated the microcystins (MCs) removal of 85% at a dose of 60 mg/L. On the other hand, Sun et al. [19] claimed that conventional poly aluminum chloride (PAC) is ineffective in removing MCs from wastewater. In addition, most of the Ti-based coagulation studies are performed on algae cultured synthetic wastewater; hence, an evaluation of Ti-based coagulation in real algae bloomed surface water would be a significant addition. Besides, the Ti-based flocculated sludge can be recycled to produce highly valuable titania ($TiO_2$) nanoparticle (NPs) [15,20,21]. It has been reported that around 446.5 kg $TiO_2$ NPs can be generated from a water treatment facility, having a treatment capacity of 25,000 $m^3$/d [15,18]. Additionally, $TiO_2$ NPs from Ti-based flocculated sludge are found to be morphologically superior with equivalent photocatalytic activities as compared to commercially available P25 [22]. $TiO_2$ is a widely applied compound in the photocatalytic and photovoltaic sector, due to its unique optical and electronic properties [23,24]. Although the use of photocatalyst in the degradation of gaseous pollutants is not yet a well-accepted practice, due to the low effectiveness and release of toxic intermediates [25,26], the photocatalytic application of $TiO_2$ in the field of air purification has received significant attention. Mostly, the photodegradation of atmospheric nitrogen oxide ($NO_x$) using $TiO_2$ incorporated substrate showed very encouraging outcomes [27,28]. The reports indicated that even a very low short-term concentration of NO (0.05 ppm to 0.2 ppm) could cause respiratory problems, such as asthma and bronchitis [29,30]. Moreover, $NO_x$ could cause the formation of photochemical smog and tropospheric ozone, along with the hazardous phenomena, like acid rain [31]. Hence, the abatement of $NO_x$ using photocatalytic $TiO_2$ bears immense significance.

$TiO_2$ is an n-type semiconductor, whose conduction band (CB) and valence band (VB) are at an energy level of −0.03 eV and 2.9 eV, respectively, which results in the band energy gap of 3.2 eV [23,32]. Hence, during the solar irradiation of $TiO_2$, when the energy of the incident photon ($E_p$) is equivalent to the band energy gap ($E_{gap}$), it gets absorbed. This phenomenon leads to the formation of hydroxyl (OH•) and superoxide (•$O_2$) radicals by utilising surrounding $O_2$ and $H_2O$, respectively [24]. Finally,

the active $OH\bullet$ and $\bullet O_2$ oxidizes the adsorbed $NO_x$ into nitrate ($NO_3^-$) ions. Figure 1 illustrates a schematic of the whole process [28,33]. The photodegradation of $NO_x$ has been exclusively studied while using $TiO_2$ blended cementitious materials in both laboratory and pilot scale setup [34]. Moreover, a number of real life applications of $TiO_2$ blended substrate for atmospheric $NO_x$ removal showed convincing results [35,36]. Photocatalytic ($TiO_2$ mixed) pavement blocks are found to effectively remove $NO_x$ while using UV irradiance above the light intensity of 1 W/m$^2$, and more than 80% $NO_x$ removal was reported at an initial NO concertation of 0.05 ppm to 1.0 ppm [35]. One of the recent studies [37] evaluated the $NO_x$ removal of sprayed $TiO_2$ on the retaining wall of an expressway (Gyeongbu, Korea) and it reported a maximum average daily removal of 22%.

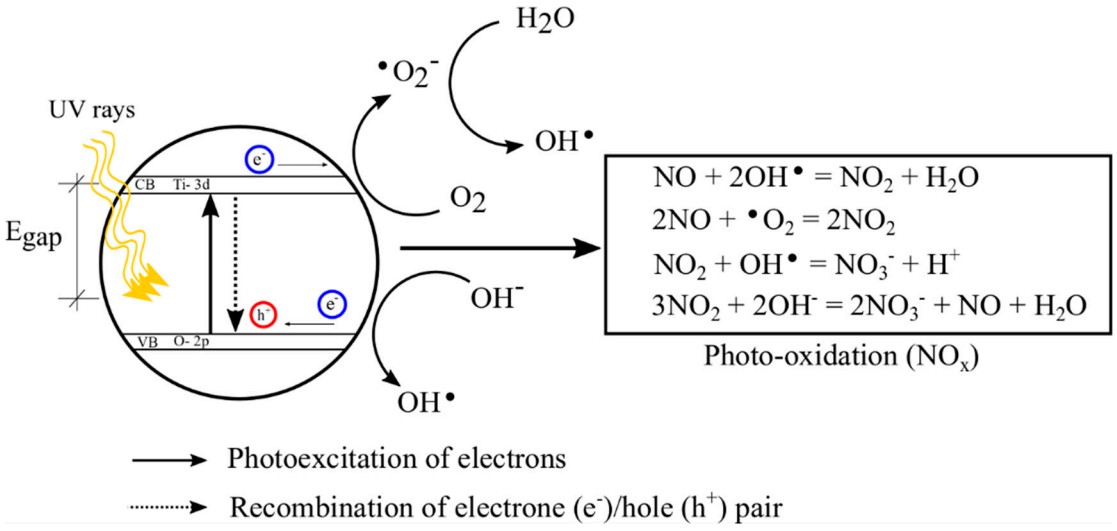

**Figure 1.** Nitrogen oxide ($NO_x$) removal mechanism using titania ($TiO_2$).

The conventional $TiO_2$ production processes are not very environmentally friendly, and they generate hazardous wastes, requiring advanced management [22]. Hence, an alternative $TiO_2$ production scheme, which is economic and environmentally friendly, is always appreciated. The novelty of the present work is the preparation of photocatalytic anatase $TiO_2$ from flocculated algal sludge to remove gaseous $CH_3CHO$ and $NO_x$. Consequently, the flocculation efficiency of $TiCl_4$ was compared with the commercially available PAC in algae bloomed surface water. Later, $TiO_2$ NPs were recovered from algae flocculated sludge, and the extent of photodegradation of prepared $TiO_2$ was compared with P25.

## 2. Results and Discussion

### 2.1. Coagulation Efficiency of TiCl$_4$ Comparing PAC

The coagulation efficiency of the $TiCl_4$ was assessed in comparison with commercially available PAC in algae bloomed lake water. The coagulant doses were varied from 0.1 to 0.3 g/L and coagulation efficiency was measured based on turbidity, COD and TP. Table 1 shows the variations in coagulation efficiency, along with the standard errors for the utilised coagulants. Irrespective of the coagulants, coagulation efficiencies increased with continuous increments in coagulant doses. Interestingly, $TiCl_4$ was found to be superior in the removal of turbidity when compared to PAC at an equal amount of coagulant doses. Using 0.1 g Ti/L, approximately 85% turbidity and 70% of TP were removed. Whereas, 0.1 g Al/L only removed 42% and 50% of turbidity and TP, respectively. Both of the coagulants were found to be inefficient in removing COD from algae bloomed water at a dose of 0.1 g/L. In comparison with other studies on the treatment of algae bloomed water [8,17] the coagulant doses that were required in the present work were very high, mostly because of the variation in feed water quality. Additionally, the extent of turbidity, COD and TP were several folds

higher when compared to all of the cited studies. For instance, Xu et al. [14] analysed the coagulation efficiency of $TiCl_4$ and PAC, and reported an optimum dose of 0.05 g Ti/L and 0.03 g Al/L, respectively. They found that, at an optimum dose, PAC performed better in removing turbidity as compared to $TiCl_4$, which is in contradiction with the current study. This could be due to the lower basicity of the utilised PAC in the current study, leading to the formation of less amount of positively charged $Al(OH)_n^{(3-n)+}$ hydrolyzates [38,39]. Besides, the number of valence electrons in Ti is higher than that of Al and, upon hydrolysis, vigorously generates highly charged positive hydrolyzates ($Ti(OH)_n^{(4-n)+}$), attributing to elevated charge neutralization capacity [14,40].

**Table 1.** Turbidity, COD and TP results with different concentrations of $TiCl_4$ (as Ti) and poly aluminum chloride (PAC) (as Al) in algal turbid water.

| Dosage (g/L) | Ti | | | Al | | |
|---|---|---|---|---|---|---|
| | Turbidity (NTU) | COD (mg/L) | TP (mg/L) | Turbidity (NTU) | COD (mg/L) | TP (mg/L) |
| 0.0 | 500 ± 0.018 | 117 ± 0.503 | 3.61 ± 0.261 | 500 ± 0.018 | 117 ± 0.503 | 3.61 ± 0.261 |
| 0.1 | 75 ± 0.050 | 115 ± 0.362 | 1.08 ± 0.251 | 290 ± 0.052 | 113 ± 0.233 | 1.81 ± 0.050 |
| 0.2 | 1.9 ± 0.178 | 115 ± 0.308 | 0.59 ± 0.122 | 65 ± 0.051 | 84 ± 0.328 | 0.32 ± 0.102 |
| 0.3 | 1.0 ± 0.156 | 39 ± 0.135 | 0.11 ± 0.051 | 21 ± 0.044 | 86 ± 0.244 | 0.16 ± 0.057 |

Using Table 1, the successive increase in coagulant dose up to 0.3 g/L as Ti and Al showed a continuous improvement of coagulation efficiency. At a dose of 0.3 g Ti/L, the turbidity, COD, and TP removal were 99.8%, 66.7%, and 96.9%, respectively. Alternatively, at a similar dose, PAC showed equivalent turbidity (95.8%) and TP (95.5%) removal, but the COD removal was 40.2% lower (see Table 1). In general, coagulation using PAC is suitable for removing high molecular weight organic matter with hydrophobic nature [8]. However, PAC was found to be inefficient in removing uncharged and hydrophilic biopolymers (i.e., polysaccharides) [19]. On the other hand, $TiCl_4$ was found to be efficient in removing organic matters with a very low molecular weight (<350 Dalton) and it can remove uncharged biopolymers to some extent through sweep flocculation and the bridging of formed complexes with AOM [18]. A previous report indicated that, at 0.06 g Ti/L of coagulant, $TiCl_4$ could remove 85% of algae originated microcystins [14]. From the COD removal values that are tabulated in Table 1, it can be expected that the AOM in the collected lake water contained a large portion of LMW organic compounds. Hence, in this study, $TiCl_4$ was found to be more efficient in abating COD as compared to PAC.

Furthermore, high doses of the coagulants may have degraded the algae cell membrane causing the release of IOMs, making it hard to reduce COD [2]. Presumably, $TiCl_4$ coagulation in this study might have caused algal cell damage, thus, COD remained approximately the same, while the coagulant dose was increased from 0.1 g Ti/L to 0.2 g Ti/L. Although the zeta potentials of the flocculated flocs were not measured in this study, previous studies using optimum Ti dose for coagulation of synthetic water containing AOM indicated that the zeta potential remained within the range of −5.0 to 0.0 mV [17,18]. To be exact, Chekli et al. [18] calculated the zeta potential at optimum dose of 0.05 g Ti/L, and found the optimum zeta potential to be −2.03 mV. The coagulation mechanisms of $TiCl_4$ in algae bloomed water could be attributed to charge neutralization, sweep flocculation, and bridging of developed Ti hydrolyzates incorporated AOM complexes [14,41]. Another prominent advantage of $TiCl_4$ coagulation in this study was the larger floc size when compared to PAC, which was even detectable with bare eyes. The larger and more compact flocs during the Ti-based flocculation are mainly due to the large atomic radius and a high number of valence electrons [15,16,22].

$TiCl_4$ has been found very efficient in removing phosphorous from wastewater. Jeon et al. [42] reported that both Ti and Al-based coagulants perform similarly when the initial pH of the feed water is 7, which is in agreement with the current study. At an alkalinity level of 100 mg $CaCO_3$/L, they have reported 99% phosphorous removal using a coagulant dose of 0.02 g Ti/L and 0.01 g Al/L, respectively.

It was reported that an increase in alkalinity of feed water was responsible for the requirement of elevated coagulant doses [42]. Consequently, in the current study, the presence of algae in feed water might have increased the alkalinity, causing an augmented demand of coagulants to remove TP.

### 2.2. Physicochemical Properties of the Prepared A-TiO$_2$

The physicochemical properties of the prepared algae bloomed water treated TiO$_2$ (A-TiO$_2$) NPs were evaluated via scanning electron microscope/energy dispersive X-ray spectroscope, X-ray diffraction (SEM/EDX, XRD), and physisorption of N$_2$. The prevailing physicochemical properties were also compared with annealed algal sludge (A-Residue) from the collected raw lake water sample. Figure 2 shows the SEM image of A-Residue and A-TiO$_2$. The particles that were present in A-TiO$_2$ were well dispersed, and the average particle size was between 20–30 nm (see Figure 2b). On the other hand, particles present in A-Residue were partially agglomerated, and the average particle size was approximately 50 nm. The particles of the prepared A-TiO$_2$ corresponds well with the TiO$_2$ NPs that were prepared by Park et al. [43] and El Saliby et al. [44] through flocculation of dye wastewater and biologically treated sewage effluent, respectively, while using Ti-based coagulant.

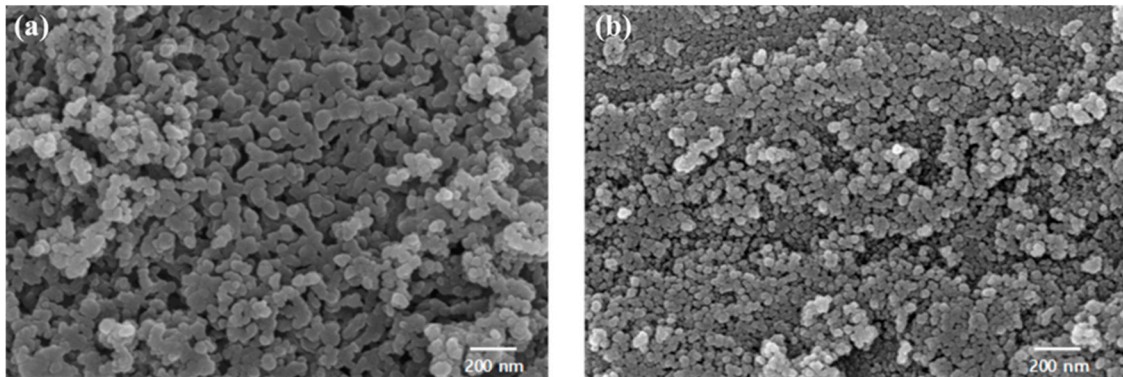

**Figure 2.** SEM images of (**a**) A-Residue; and, (**b**) A-TiO$_2$.

In contrast, TiO$_2$ prepared from secondary sewage effluent and paper mill wastewater showed a smaller (<20 nm) particle size [21,22]. It has been reported that the particle size of TiO$_2$ NPs developed through Ti-based flocculation primarily depends on the quality of the feed water that has been used [6,45,46]. Moreover, the substitutional and interstitial doping of impurities from the feed water can significantly affect the particle size of the prepared TiO$_2$ NPs [24,47]. For instance, the high Ca (large atomic radius when comparing Ti) content (see Table 2) in A-TiO$_2$ crystal might have contributed towards the enhancement of crystal size through substitutional doping [32].

**Table 2.** Energy dispersive X-ray spectroscope (EDX) and Brunauer-Emmett-Teller (BET) surface area results of A-Residue and A-TiO$_2$.

| Material | Weight (%) | | | | | | | | | | | | S$_{BET}$ (m$^2$/g) |
| --- | --- | --- | --- | --- | --- | --- | --- | --- | --- | --- | --- | --- | --- |
| | O | Na | Mg | Al | Si | P | S | Cl | K | Ca | Fe | Ti | |
| A-Residue | 45.5 | 2.0 | 3.8 | 4.4 | 13.7 | 3.5 | 3.1 | 2.4 | 5.0 | 11.1 | 5.5 | - | 9 |
| A-TiO$_2$ | 49.3 | 1.9 | 1.3 | - | 0.6 | - | 1.1 | 2.7 | 0.6 | 3.3 | - | 39.2 | 40 |

Table 2 demonstrates the elemental composition of the prepared A-Residue and A-TiO$_2$ from EDX analysis along with the estimated BET specific surface area (S$_{BET}$). A noticeable increase (approximately four times) in S$_{BET}$ has been observed in A-TiO$_2$ compared to A-Residue, which support the obtained results of smaller particle size of A-TiO$_2$ presented in Figure 2. Using Table 2, various elements were found in the analysed A-Residue with a notable amount of Si (13.7%), Ca (11.1%), Fe (5.5%), and O (45.5%). Possibly, the prevailing AOMs (EOMs and IOMs) of the feed water were the sources of the

elements that were observed in A-Residue. In general, AOMs contain biopolymers (i.e., polysaccharides and proteins) (38.5%), humic substances (9.9%), building blocks (27.1%), and low molecular weight organics (i.e., alcohols, aldehydes, ketones, and monoprotic organic acids) (23.9%), thus possibly contributing to impurities on A-Residue [18]. Surprisingly, certain amounts of Cl were found in both A-Residue and A-TiO$_2$, which could have been attributed from the existing residual chlorine of the lake water. Nevertheless, the fraction of impurities was reduced in the prepared A-TiO$_2$ NPs (see Table 2), as Ti hydrolyzates develop chemical complexes [14] with existing AOM and at 600 °C, anatase TiO$_2$ is the dominant crystal structure [21].

The S$_{BET}$ of the prepared A-TiO$_2$ was approximately similar to that of commercially available P25 (42.3 m$^2$/g) despite the presence of varying impurities [15]. However, the S$_{BET}$ of the A-TiO$_2$ NPs were found to be less as compared to the TiO$_2$ NPs that were prepared from Ti-based flocculation of drinking water (90.2 m$^2$/g) [22], secondary sewage effluent (103.5 m$^2$/g) [22], seawater (68.1 m$^2$/g) [48], and dye wastewater (76 m$^2$/g) [20]. Substantial doping of the impurities in A-TiO$_2$ crystal might have been the possible reason of the reduced specific surface area [23], which is evident from the reduced fraction of Ti (39.2%) in relation to O (49.3%) (see Table 2).

The X-ray diffraction patterns of A-Residue and A-TiO$_2$ are compared in Figure 3a, and the result bears convincing resemblance with the elemental composition that is found in EDX analysis (see Table 2). In A-Residue, SiO$_2$ crystal diffraction peaks at 20.9°, 26.7°, and 36.6°, and CaO crystal diffraction peak at 50.2° were observed. Alternatively, A-TiO$_2$ mostly showed anatase TiO$_2$ crystal peaks at 25.3°, 37.9°, and 47.9°, which is in agreement with a reported study that the incineration of Ti flocculated sludge at 600 °C generates photocatalytically active anatase TiO$_2$ [21]. An insignificant diffraction peak for SiO$_2$ was observed at 26.7°; however, it was presumed to be primarily present in an amorphous state. Furthermore, Figure 3a depicts the presence of doped pollutants (substantially/interstitially) in A-TiO$_2$, which inhibited the crystalline diffraction intensity of anatase TiO$_2$ [22].

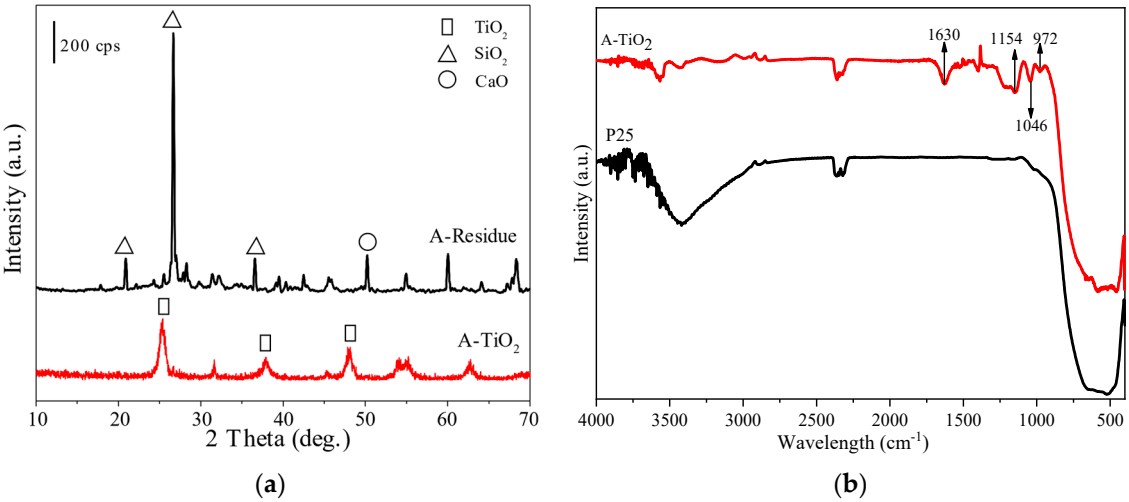

**Figure 3.** (**a**) X-ray diffraction (XRD) patterns, (**b**) Fourier transform infrared (FT-IR) spectra of A-Residue and A-TiO$_2$ prepared from algal turbid water.

The FT-IR spectra of commercially available P25 and A-TiO$_2$ are illustrated in Figure 3b. For both A-TiO$_2$ and P25, a range of absorption broad peak was observed in between 400 cm$^{-1}$ to 900 cm$^{-1}$, which is characteristics spectra for TiO$_2$ representing the vibration of Ti-O-Ti [46]. Another broad absorption band around 3435 cm$^{-1}$ in P25 indicated the characteristic vibration mode of Ti-OH [47], could not be observed in A-TiO$_2$. However, the absorption peak around 1630 cm$^{-1}$ in A-TiO$_2$ indicated the presence of hydroxyl group [46,48]. The shift vibration peaks around 1154 cm$^{-1}$, 1046 cm$^{-1}$ and 972 cm$^{-1}$ in A-TiO$_2$, represented the presence of C-O-C, C-O, and Si-O-Si in lower intensity, which exhibited the presence of C and Si on prepared A-TiO$_2$ [46,49]. Several previous studies

concerning sludge generated TiO$_2$ showed the traces of C, which was found to be beneficial during photocatalysis due to tailored energy bandgap from doping effect. A thorough quantitative study on the X-ray photoelectron spectroscopy (XPS) spectra of A-TiO$_2$ and P25 was conducted to compare the surface chemical compositions. Figure 4a illustrates the survey spectra of the as-prepared photocatalysts. The survey spectra delineate the presence of Ti 2p and O 1s region, thus confirming the formation of TiO$_2$ from the prescribed methodology in this manuscript [46]. Using high-resolution XPS, the Ti 2p and O 1s region were further assessed, and the dominant peaks were deconvoluted using Lorentz line fit and presented in Figure 4b,c respectively. The Ti 2p spectra of A-TiO$_2$ showed two characteristic peaks around 459.39 eV and 465.07 eV. Whereas, P25 depicted the Ti2p characteristic peaks around the binding energies of 458.41 eV and 464.31 eV, which can be attributed to Ti 2p$_{3/2}$ and Ti 2p$_{1/2}$, respectively, of Ti$^{4+}$ [50]. A major shift (465.07 eV) in the Ti 2p$_{3/2}$ peak of A-TiO$_2$ was observed as compared to the P25, which can indicate the generation of Ti$^{3+}$ as the local trap state [51]. Such local trap states can restrict the recombination rate of photogenerated e$^-$/h$^+$ pairs along with the narrowing of energy bandgaps [24].

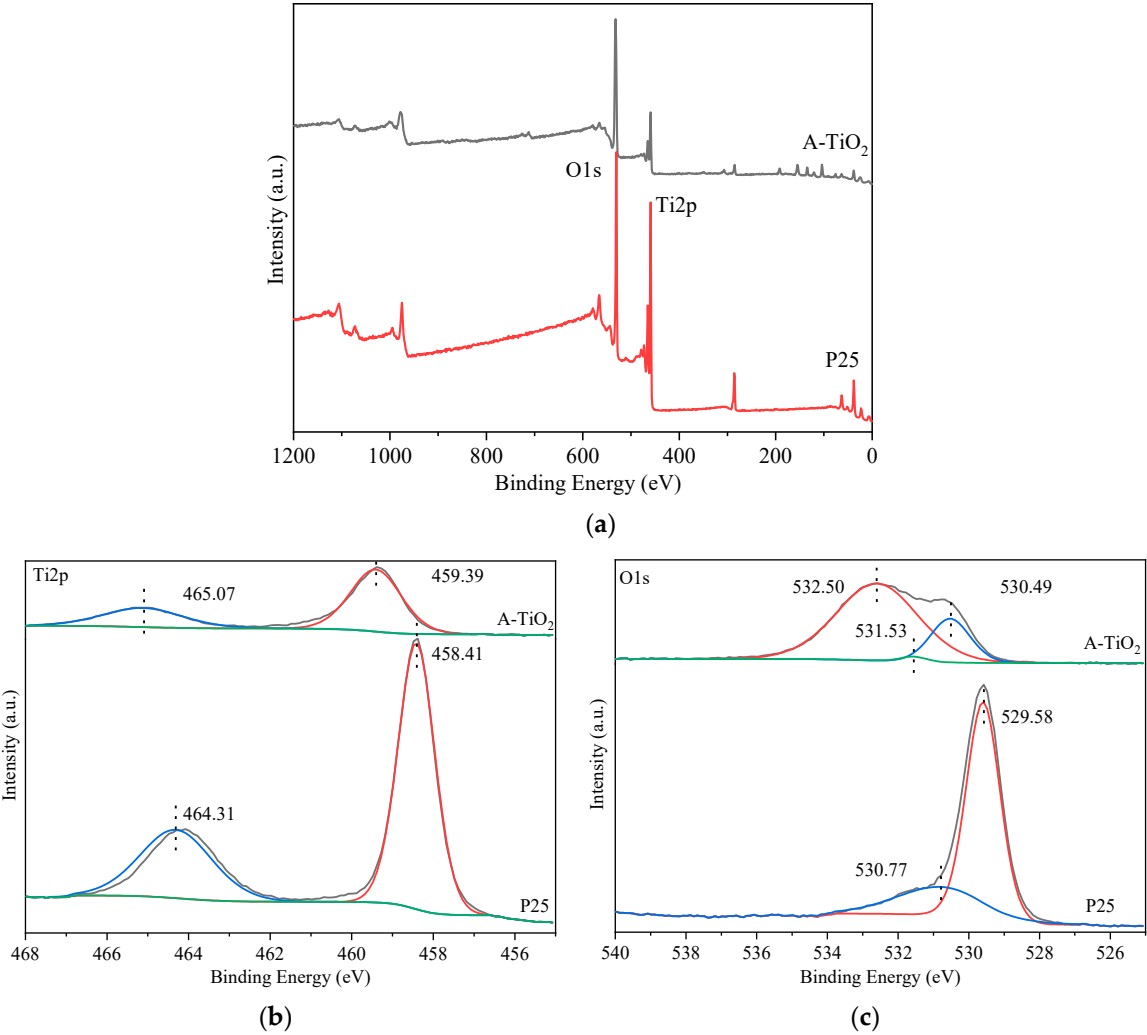

**Figure 4.** (**a**) X-ray photoelectron spectroscopy (XPS) survey spectra, (**b**) Ti 2p spectra, and (**c**) O1s spectra of A-TiO$_2$ and P25.

Although this study did not report any visible light photocatalysis experimental results, the narrowed energy bandgap of the as-prepared A-TiO$_2$ could mean visible light activation. Moreover, the EDX results that are presented in Table 2 show the presence of Ca in the chemical composition of A-TiO$_2$, and the shift of Ti 2p towards higher binding energy could be due to the presence of dopant

in A-TiO$_2$. Similar changes in chemical compositions were observed in some of the relevant studies. For instance, Shon et al. [15] performed XPS analyses on sludge generated TiO$_2$ from the coagulation of synthetic wastewater to determine the composition of the TiO$_2$, and to classify the valence states of different atoms present. They have recorded the sample spectrum for Ti, O, C, and P while using multi-scan recording mode, which revealed that the atomic percentages were 26.9%, 51.5%, 15.8%, and 5.8%, respectively. In addition, they have the Ti2 p line's binding energy at about 458.96 eV, confirming Ti was mainly present as Ti$^{4+}$. This phenomenon suggests that a higher degree of oxidation occurred during the calcination of Ti incorporated sludge. Whereas, following a similar scheme used in this manuscript and Shon et al. [15], Mian and Liu [46] prepared TiO$_2$ by sewage sludge coagulation and stated that the Ti 2p line was moved to higher binding energy as 459.1 eV. Accordingly, they concluded that sludge that was produced TiO$_2$ could facilitate the development of local trap state Ti$^{3+}$ and increase the photocatalytic efficiency. The O 1s spectra of A-TiO$_2$ illustrated in Figure 4c showed characteristic peaks of anatase TiO$_2$ around 530.49 eV and 531.53 eV, which can be dedicated to the Ti-O and Ti-OH bond, respectively [51]. Presumably, due to the presence of trace amounts of dopants, the O 1s peaks shifted towards higher binding energies compared to P25. Additionally, the peak around 532.50 eV is a characteristic peak of the C=O bond, which might have resulted from the impurities present in the as-prepared A-TiO$_2$ [51].

### 2.3. Photocatalytic Activity of Prepared A-TiO$_2$ NPs

The photocatalytic activities of the developed A-TiO$_2$ NPs were assessed under UV irradiation through the photodegradation of gaseous acetaldehyde (Figure 5) and NO$_x$ (Figure 6). Additionally, the degree of photoactivity was compared with commercially available P25. In the case of acetaldehyde decomposition under UV irradiation, to exclude the losses to photodecomposition along with other heterogenous losses, i.e., uptake and hydrolysis, blank experiments were initially performed with 2000 ppmv of acetaldehyde gas mixture in the reactor under UV-A irradiation without any presence of photocatalysts. During UV irradiation for over 200 min., no such observable losses or systemic patterns were observed. Acetaldehyde was adsorbed on A-TiO$_2$ and P25 for 80 min. in dark condition (no UV irradiation) and, at the initiation of UV-A lamp, the concentration of acetaldehyde was found as approximately 1820 ppmv for both A-TiO$_2$ and P25 (see Figure 5). Using Figure 5, under UV irradiation at the end of 200 min. A-TiO$_2$ and P25 removed almost 85.7% and 96.7% of the imparted acetaldehyde gas, respectively. Past studies have shown that the TiO$_2$ photocatalytic degradation rate adopted the traditional pseudo-first-order kinetics pattern of Langmuir–Hinshelwood [52]. The kinetic equation can be expressed as ln C/C$_0$ = −kt. Where, k is the pseudo-first-order reaction rate constant (min$^{-1}$), C is the concentration at reaction time t, and C$_0$ is the initial concertation. In Figure 5b, the value k was calculated from the slope, where ln C/C$_0$ versus t was plotted. Thus, the reaction rate constant of A-TiO$_2$ and P25 were found as 0.0169 min$^{-1}$ and 0.0311 min.$^{-1}$, respectively.

Similarly, following ISO 22197-1, the NO$_x$ removal efficiency of A-TiO$_2$ was compared with P25 (Figure 6). At the beginning of UV irradiation, A-TiO$_2$ and P25 both exhibited a rapid reduction of NO, probably due to the combined effect of adsorption and photodegradation [43]. Moreover, the maximum removal of NO achieved under UV irradiation was 57.10% and 78.22% for A-TiO$_2$ and P25, respectively. Apparently, the HNO$_3$ developed during photooxidation of NO might have adsorbed on the surface of TiO$_2$ and diminished the level of photodegradation [30]. Hence, both A-TiO$_2$ and P25 indicated consecutive increase in concentration of NO. After 60 min. of UV irradiation, the NO removal efficiency of A-TiO$_2$ and P25 were determined as 16% and 50%, respectively. In the continuous flow reactor, the average initial concentration of NO was found as 6.95 µmol and 7.01 µmol for A-TiO$_2$ and P25, respectively, as illustrated in Table 3. Under UV irradiation for 60 min., the prepared A-TiO$_2$ removed 1.95 µmol of NO on average, which is approximately 27.99% of the initial concentration. Whereas, the used P25 showed an average removal of 54.17% for the same duration of UV exposure. Meanwhile, under UV light, the average NO$_2$ concentration was increased by 16.50% and 19.79% for A-TiO$_2$ and P25, respectively, which shows a higher selectivity of A-TiO$_2$ towards NO$_2$ production.

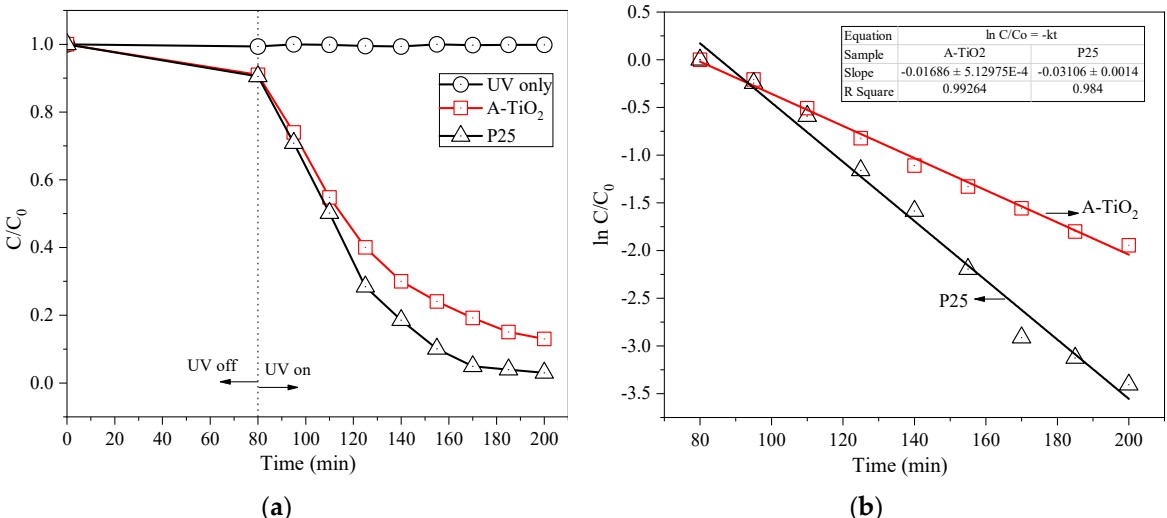

(**a**)  (**b**)

**Figure 5.** (**a**) Removal rate of acetaldehyde and, (**b**) first-order kinetics under UV irradiation over A-TiO$_2$ and P25 NPs.

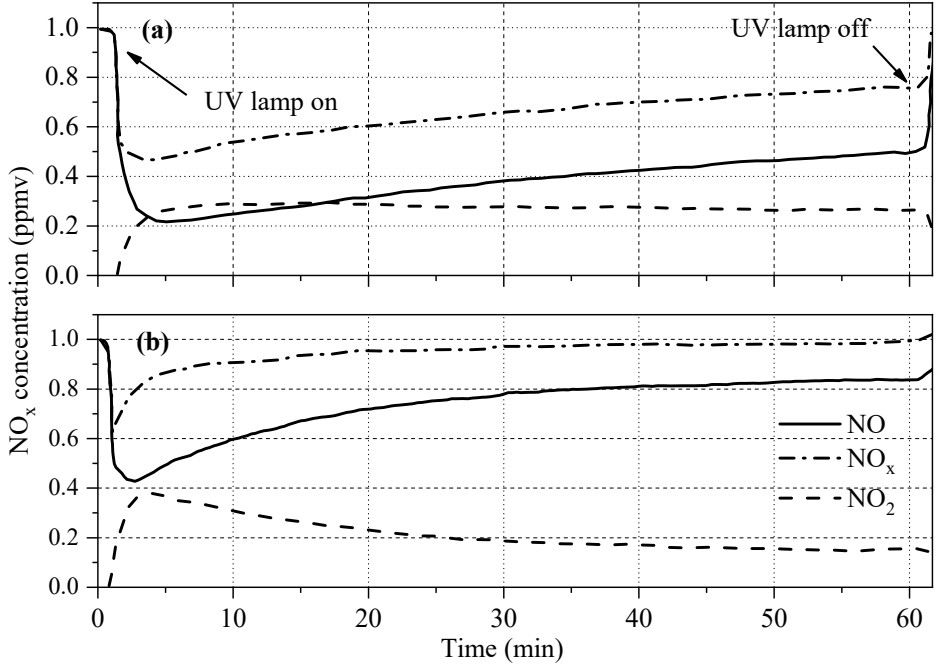

**Figure 6.** Removal of NO$_x$ under UV irradiation over (**a**) P25 and (**b**) A-TiO$_2$ NPs.

**Table 3.** Average NO removal and NO$_2$ generation rates for the prepared samples.

| Sample | NO$_{(initial)}$ | NO$_{(removal)}$ | %NO$_{(removal)}$ | NO$_{2(initial)}$ | NO$_{2(production)}$ | %NO$_{2(production)}$ |
|---|---|---|---|---|---|---|
| - | μmol | μmol | - | μmol | μmol | - |
| NPs | | | | | | |
| A-TiO2 | 6.95 | 1.95 | 27.99% | 0.45 | 1.15 | 16.50% |
| P25 | 7.01 | 3.80 | 54.17% | 0.65 | 1.39 | 19.79% |
| Mortar blocks | | | | | | |
| A-TiO2 (5%) | 7.46 | 0.24 | 3.24% | 0.03 | 0.08 | 1.08% |
| A-TiO2 (10%) | 7.49 | 0.46 | 6.14% | 0.01 | 0.13 | 1.78% |
| P25 (5%) | 7.45 | 0.45 | 6.02% | 0.02 | 0.14 | 1.88% |

From Figures 5 and 6, it is clear that the prepared A-TiO$_2$ was found to be inferior to photodegrade acetaldehyde and NO$_x$ as compared with P25. The aforementioned is in contrary with most of the previous studies, where TiO$_2$ NPs produced from Ti-based flocculation were found to be either superior or almost similar to P25 with regards to the photodegradation of volatile organic compounds [6,43,44]. The following postulates may explain the reduced photocatalytic activities of the prepared A-TiO$_2$:

- Firstly, the apparent density of A-TiO$_2$ was estimated as 1.2 g/mL, which is approximately six times larger than that of P25 (0.19 g/mL), causing a small number of adsorption sites for A-TiO$_2$. However, when considering the substantial difference in the available adsorption sites, the reduction in maximum removal of acetaldehyde and NO$_x$ was only 11% and 20% respectively, when comparing A-TiO$_2$ to P25 NPs.

- Secondly, it is evident that doped metals and nonmetal significantly affect the photo activity of a photocatalyst [23]. The EDX analysis of the current study showed notable amounts of Ca (3.3%) and Cl (2.7%) in A-TiO$_2$. Castro and Durán [53] reported that Ca doping on TiO$_2$ at a very low concentration (<3 wt.%) could reduce the band energy gap of TiO$_2$ and enhance the photodegradation of methyl orange under solar irradiation. However, the doped Ca of substantial amount can act as recombination site for e$^-$/h$^+$ pair generated during photodegradation (see Figure 1) and might cause a reduced level of photoactivity [32]. Similarly, Wang et al. [54] illustrated that a certain amount (2 atomic%–4 atomic%) of Cl as the dopant in TiO$_2$ could activate TiO$_2$ under visible light, and the extent of light absorption within the UV range can get reduced, so as the photoactivity. Hence, in the current study, the dual effect of the doped Ca and Cl might have reduced the photocatalytic activity of A-TiO$_2$.

*2.4. Photocatalytic Activity of Prepared A-TiO2 NPs*

Figure 7 depicts the NO$_x$ removal performance of photocatalytic mortars prepared by adding 5 wt.% and 10 wt.% of A-TiO$_2$. The results were compared with mortar prepared using 5 wt.% of P25. Under UV irradiation, the NO concentration in the reactor gradually reduced and stabilized approximately after 5 min., regardless of the mortar used. The UV-A lamp remained turned on for 60 min., and the stabilized NO concentration was found constant throughout this time irrespective of the employed photocatalytic mortar. In the photocatalytic mortar containing 5 wt.% and 10 wt.% of A-TiO$_2$, the NO concentration was stabilized at 0.96 ppmv and 0.93 ppmv, respectively.

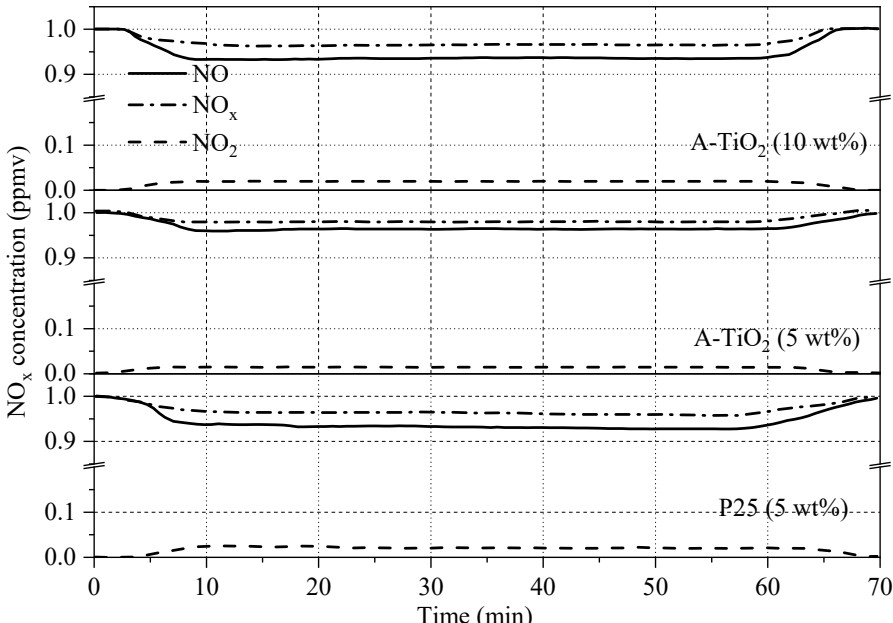

**Figure 7.** NO$_x$ removal performance of the manufactured photocatalytic mortars under UV-A lamp.

Meanwhile, the maximum removal of NO using P25 (5 wt.%) mixed mortar was 0.93 ppmv. $NO_2$ production rate was found to be proportional to the NO removal rate in both A-$TiO_2$ and P25 mixed mortar, as exhibited in Figure 7. The average $NO_2$ concentration in the reactor increased from 0.03 μmol to 0.08 μmol, 0.01 μmol to 0.13 μmol, and 0.02 μmol to 0.14 μmol for mortar blocks containing A-$TiO_2$ (5%), A-$TiO_2$ (10%), and P25 (5%), respectively, under UV irradiation of 70 min., as shown in Table 3.

At equal wt.%, A-$TiO_2$ showed 50% less NO removal efficiency when compared to P25. The results in Figures 5 and 6 indicated a significant difference in $NO_x$ removal behavior for photocatalytic (A-$TiO_2$ or P25) NPs and photocatalyst (A-$TiO_2$ or P25) mixed mortar. The corresponding NPs under continuous UV light showed a decreasing trend (see Figure 6) during NO removal, which was not the scenario in the case of the prepared mortar blocks. Possibly, the synergistic effects of the cementitious materials on $TiO_2$ NPs were the prevailing reasons [35]. It has been reported that the final product of NO removal using $TiO_2$ is $NO_3^-$ (see Figure 1), which can react with Ca in surrounding hydrated cement and produce $Ca(NO_3)_2$ rather than inhibiting the photocatalytic activity of $TiO_2$ through adsorption on photoactive sites [28,30] or the regeneration of NO under UV [55]. Hence, the photodegradation of NO remained almost constant under the UV lamp.

As mentioned before, the apparent density of A-$TiO_2$ was significantly higher than that of P25, which leads to its poorer NO abatement rate than P25. Despite the notable difference (almost six times) in density, the extent of photoactivity only reduced by 50%. It is presumed that complex interactions between cement and A-$TiO_2$ NPs, along with physical properties of mortar contributed towards the improved removal of NO [37,50]. Park et al. [43] prepared $TiO_2$ from flocculation of dye wastewater and their $TiO_2$ performed at par with commercial P25 for the removal of atmospheric $NO_x$, since the apparent density of their $TiO_2$ was not very high when compared with P25. Therefore, an improvement in the physical properties of A-$TiO_2$ developed in the current study could further enhance the extent of photoactivity. Moreover, from Figure 7, it is evident that A-$TiO_2$ at 10 wt.% in mortar performed similarly as with P25 (5 wt.%). The A-$TiO_2$ used in this study was prepared from recycling sludge, so it is considerably cheaper when compared to P25. Therefore, the problem concerning high apparent density can be counterbalanced by increasing the amount of A-$TiO_2$ used.

## 3. Materials and Methods

### 3.1. Chemical Reagents and Simulated Algal Wastewater

Algae bloomed water was collected from Daecheongho Lake, which is one of the largest manmade lakes in South Korea and it fulfills various water demand of Daejeon and Cheongju. The raw lake water was diluted using tap water, and the pH was adjusted using 1.0 N NaOH (>97% purity; Daejung Chemicals and Metals, Siheung-si, South Korea). The physicochemical properties of the simulated water were determined, as follows: pH 7–7.5, temperature 22–25 °C, turbidity 500 NTU, COD 117 mg/L, and TP 3.61 mg/L. In this study, the flocculation efficiency of $TiCl_4$ is compared with commercially available PAC. The stock solution of $TiCl_4$ was prepared while using concentrated $TiCl_4$ (>99% purity, density 1.73 g/mL; Sigma Aldrich, Castle Hill, NSW, Australia). A 20 wt.% $TiCl_4$ solution was prepared by adding (dropwise) 46.4 mL of concentrated $TiCl_4$ to a predetermined volume (400 mL) of DI water (frozen cubes) under continuous stirring. Industry grade PAC was directly utilised, having 10.2 wt.% of $Al_2O_3$ and 64% basicity.

### 3.2. Jar Tests

Standard jar tests were conducted to compare the coagulation efficiencies of $TiCl_4$ and PAC using a programmable jar tester (PB-900TM, Phipps and Bird, USA). The simulated algal wastewater of 500 mL was filled in a 1.0 L beaker, and the coagulant doses (as Ti or Al) were varied from 0.1 g/L to 0.3 g/L at an increment of 0.1 g/L. The addition of coagulant was followed by rapid mixing at 100 rpm for 1 min., and a slow mixing at 20 rpm for 20 min. Finally, the samples were settled for 20 min.

before analysing the target physicochemical properties of the effluent. The coagulation efficiency of the coagulants were compared from the perception of residual turbidity, COD, and TP. The sample water after coagulation was collected from 2.0 cm below the water surface in the beaker, and turbidity was determined immediately while using turbidity meter. Besides, COD and TP were determined using UV/VIS spectrophotometry, utilising potassium permanganate ($KMnO_4$) as an oxidizing agent. All of the experiments were conducted in triplicates, and the average values with standard errors are reported.

### 3.3. Preparation and Characterization of $TiO_2$ from Flocculated Algal Sludge

Following the jar tests, Ti-based algal flocculated sludge was collected, and titania ($TiO_2$) nanoparticles (NPs) were prepared following the protocols described in the literature [15,16], and they are illustrated in Figure 8. The prepared A-$TiO_2$ NPs were characterized using a scanning electron microscope that was associated with energy dispersive X-ray spectroscope (SEM/EDX, Rigaku, Wilmington, MA, USA). The specific surface area of the prepared A-$TiO_2$ was determined using physisorption at 77K that was accompanied with $N_2$ gas (ASAP 2020, Micromeritics, Norcross, GA, USA). The multi-point Brunauer-Emmett-Teller (BET) method was adopted to calculate the specific surface area using isotherm data from $P/P_0 = 0.058$ to $P/P_0 = 0.188$.

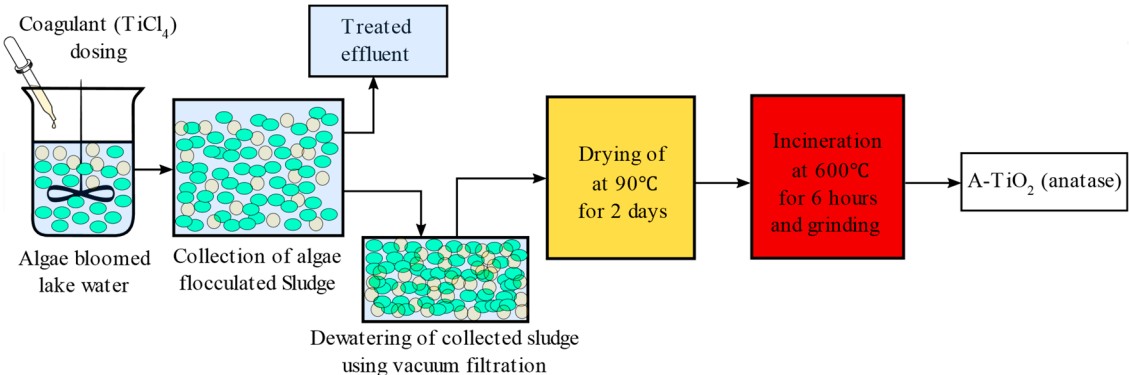

**Figure 8.** Preparation of A-$TiO_2$ using Ti flocculated algal sludge.

Finally, the crystalline phases of A-$TiO_2$ NPs were assessed using X-ray diffraction (XRD) analysis. The XRD patterns were generated on a MDI Jade 5.0 (MaterialsData Inc., Livemore, CA, USA) X-ray diffractometer (Rigaku Ultima III XRD) with Cu K$\alpha$ ($\lambda$ = 1.54056 Å) radiation source with Ni filter. The data were measured within the range of scattering angle $2\theta$ of 5°–80° at the rate of 2°/min. In addition, the collected algal sludge (without Ti-based flocculation) from Daecheongho Lake was incinerated at 600 °C and analysed using SEM, EDX, XRD, and physisorption (using $N_2$) to better evaluate the prepared A-$TiO_2$. Moreover, to analyse the elemental composition and surface hydroxylation, the Fourier transform infrared (FT-IR) spectra of the samples were taken with a Bruker Equinox 55/S spectrometer in the wavenumber range of 400–4000 cm$^{-1}$. The KBr pellet preparation technique was utilised for the sample preparation. X-ray photoelectron spectroscopy (XPS) data were collected in a Kratos Axis Ultra X-ray Photoelectron Spectroscopy system, using dry samples, in order to determine the elemental bonds. Binding energies were referenced to C 1s of C-C at 285 eV.

### 3.4. Preparation of A-$TiO_2$ Mixed Mortar Blocks

The removal of $NO_x$ using A-$TiO_2$ was assessed for both A-$TiO_2$ NPs and A-$TiO_2$ mixed mortar. The mortar blocks were prepared following the protocols stated in ISO 679:2009 [56]. In the prepared sample, the sand to cement ratio and cement to water ratio were maintained at 3:1 and 2:1, respectively. Mortar blocks (100 mm × 50 mm × 10 mm) were prepared incorporating 5.0 wt.% and

10.0 wt.% of A-TiO$_2$. Additionally, mortar blocks were prepared containing 5.0 wt.% of P25 to compare the results with commercially available TiO$_2$ (Degussa (Evonik) P25).

*3.5. Evaluation of Photocatalytic Activities of Prepared A-TiO$_2$*

### 3.5.1. Removal of Acetaldehyde under UV Irradiation

The photodecomposition of acetaldehyde (>99.9% purity, Fox Chemicals) was carried out under UV irradiation in order to assess the level of photocatalytic activity of the prepared A-TiO$_2$. An airtight cuboid (220 × 125 × 80 mm) reactor with a volume of 2 L and assimilated with a gas chromatograph/flame ionization detector (GC/FID) (HP5890 series II, Wilmington, NC, USA) was utilised to study the extent of photodegradation. Two 10 W, 341 nm ± 10 nm UV-A lamps (Sankyo Denki, F10T8BL, Kanagawa, Japan) were used to facilitate UV irradiation in the reactor. The reactor was equipped with three rubber openings, the first and second openings were respectively connected to acetaldehyde containing cylinder and air pump to ensure the mixing of air inside the reactor, while the third was connected to the attached gas chromatograph having super-Q PLOT capillary column (30 m × 0.52 mm) to measure the acetaldehyde concentration change. A petri-dish was used to place 0.5 g of A-TiO$_2$ in the reactor at a distance of 10 cm from the UV lamps. The photodegradation of acetaldehyde (2000 ppmv) was carried out for 200 min. at a temperature of 24 °C. The experiments were conducted under dark conditions for 80 min. with acetaldehyde gas mixtures exposed to the photocatalysts to assess the degradation rate due to adsorption and desorption. Finally, experiments with the above-mentioned acetaldehyde gas mixtures were performed to determine the rate of degradation due to UV-photocatalysis in the presence of UV-A light. The decomposition of acetaldehyde concentration versus irradiation time was followed up to 200 min. and measured every 15 min. The acetaldehyde removal efficiency using A-TiO$_2$ NPs were compared with the commercially available P25.

### 3.5.2. Removal of NO$_x$ under UV Irradiation

The removal efficiency of NO$_x$ (NO and NO$_2$) was evaluated for both A-TiO$_2$ NPs and A-TiO$_2$ combined mortar. The results in both cases were compared with that of P25. All of the samples were pretreated with UV irradiation of 10 W/m$^2$ ± 0.5 W/m$^2$ for 5 h while using two 10 W UV-A (Sanyo-denki, Japan) lamps. Following ISO 22197-1, a laboratory scale photocatalytic reactor was utilised to investigate the level of photodegradation of NO$_x$ [56]. The details of the system is described elsewhere [43]. Briefly, for placing the powder sample in the photocatalytic reactor a rectangular mould was used with a surface area of 50 cm$^2$ and 3 g of NPs (A-TiO$_2$ or P25) were pressed in the holder. Through the inlet of the reactor, a constant airflow of 3 L/m was maintained containing 1 ppmv of NO. The moisture content and temperature of the reactor were set as 50% and 25 °C, respectively. A NO$_x$ analyser (CM2041, Casella, Buffalo, NY, USA) was placed at the outlet of the reactor to measure the variation of NO$_x$ concentration throughout the experiment. All of the samples (NPs and mortar blocks) were pretreated with UV irradiation of 10 W/m$^2$ ± 0.5 W/m$^2$ for 5 h using two 10 W UV-A (Sanyo-denki, Kanagawa, Japan) lamps, NO$_x$ removal analysis. For NPs (A-TiO$_2$ or P25), 3 g of sample was compressed in the mould and placed in the test specimen holder as shown in the Figure 9. Then a stable flow of 1 ppmv NO was maintained for 30 min. without any UV irradiation. After 30 min. the UV-A lamps were switched on and variation of NO$_x$ was recorded in the NO$_x$ analyser for 60 min. Similarly, the prepared photocatalytic mortar blocks (100 mm × 50 mm × 10 mm) were placed in the test specimen holder and the variation in NO$_x$ (NO, NO$_2$) was recorded for 70 min. under UV irradiation. Once again, all of the experiments were conducted in triplicates and the average values are reported in this study.

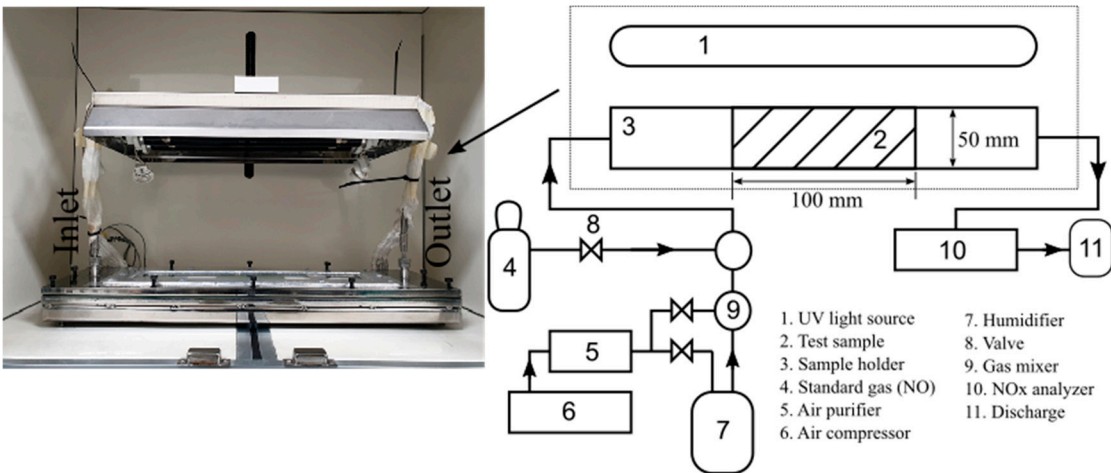

**Figure 9.** Experimental setup for the photocatalytic decomposition of NOx.

## 4. Conclusions

In the present study, TiCl$_4$ was used to conduct flocculation in algae bloomed lake water, and the accompanying jar tests showed superior turbidity, COD, and TP removal when compared with PAC. The TiO$_2$ NPs were prepared by annealing TiCl$_4$ flocculated sludge. XRD and EDX analyses confirmed the generation of anatase TiO$_2$ with some level of impurities on the annealed TiO$_2$ NPs. In addition, photocatalytic mortar blocks were prepared by mixing TiO$_2$ generated from flocculated sludge. The following are some of the important findings of the study:

- When the coagulant dose varied from 0.1 to 0.3 g/L in algae enriched wastewater, TiCl$_4$ was found to be superior in removing turbidity, COD, and TP when compared with commercially available PAC. More importantly, TiCl$_4$ removed almost 97% of the effluent TP at a coagulant dose of 0.3 g/L.
- The prepared A-TiO$_2$ NPs effectively removed 85.7% of gaseous acetaldehyde under UV-A exposure for 120 min., and by considering the pseudo 1$^{st}$ order kinetic reaction, the reaction rate constant was found as 0.0169 min$^{-1}$, which is approximately 54.34% of commercially available P25. Additionally, in a continuous flow reaction, under UV-A irradiation for 60 min., the as-prepared A-TiO$_2$ NPs was found to remove approximately 28% of NO, on average. The A-TiO$_2$ mixed mortar blocks prepared in this study showed 50% less NO$_x$ removal efficiency when compared to P25 mixed mortar blocks under UV irradiance.

Although the prepared A-TiO$_2$ showed reduced photoactivity as compared with P25, it was recovered from algal flocculated sludge, which otherwise would have required expensive disposal mechanisms for sustainable management. Hence, when considering economic factors, the novel TiO$_2$ from flocculated algal sludge can contribute towards the increasing demand of P25 in the field of air treatment.

**Author Contributions:** Conceptualization and methodology, S.M.H., H.P. and J.B.K.; formal analysis, investigation and data curation, S.M.H., H.-J.K. and I.R.; writing—original draft preparation, S.M.H.; writing—review and editing, L.T., H.-J.K., Y.-S.J. and H.K.S.; supervision, J.-H.K. and H.K.S. All authors have read and agreed to the published version of the manuscript.

**Funding:** This research was supported by a grant (18SCIP-B145909-01) from Smart Civil Infrastructure Research Program funded by Ministry of Land, Infrastructure and Transport of Korean government, and by the Technology Innovation Program (10080342, Development of Concrete Photocatalytic Finishing Plate for De-NOx) funded by the MOTIE, Korea.

**Conflicts of Interest:** The authors declare no conflict of interest.

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
