# Peer review of "Preparation and Characterization of Photoactive Anatase TiO2 from Algae Bloomed Surface Water"

_catalysts, doi:10.3390/catal10040452_

Round 1

Reviewer 1 Report

The manuscript describes waste or dirty water treatment with TiCl4 and then, catalysis of TiO2 as produced during the waste water treatment. The formed TiO2 was characterized by XRD and photocatalysis experiments with comparison with P-25. The reviewer does not know if TiCl4 is a major coagulant for waste water, however, the photocatalysis of the formed TiO2 seems that of conventional TiO2. No significance can be found on the catalysis of the produced TiO2. Thus, the reviewer thinks this manuscript should be submitted to another appropriate journal related to water treatment. 

Reviewer 2 Report

Please refer to the enclosed word file

Reviewer 3 Report

The article presents a method to first treat algae bloomed water, then extract from it TiO2 nanoparticles that can be effectively used as photocatalyst. This is a very interesting approach from a life cycle point of view, since it not only provides a purification method for water, but it also allows to produce a photocatalyst from the collected waste, which in turn can be used to sustain further purification activities.

The article is well written and the experimental setup sound and coherent. a great amount of work has been done. only one note, a better characterization of the obtained TiO2 NPs could improve the insight in the behavior observed and help improve the quality of NPs in future works - in particular, XPS/TEM analyses are suggested, to understand the level of crystallinity and in which form impurities are present in the NPs (small doping of each NP? separated particles for some of them?), plus an evaluation of the surface hydroxylation which is also very important in photocatalysis (e.g. by FTIR). 

Reviewer 4 Report

The manuscript describes the synthesis of a titanic photocatalyst from algae bloomed waters. The topic is potentially interesting because of the combined water treatment-catalyst production approach. However some minor issues must be addressed before the publication on “Catalysts”.

Issues

Rows 90-97 - The use of photo catalysis for air treatment raises it is not yet a very well accepted practice and several issues were raised both as effectiveness (e.g. [1]) and as safety (e.g. the possible emission of catalyst nanoparticles or volatile toxic intermediates). This should be at least mentioned in the introduction.

Rows 336-344 - The experimental system description should be improved with particular reference to the reactor setup (e.g. the reactor is mixed ?) and analytical sampling (e.g. sample volume and sampling system). Moreover, a zero experiment with UV and without catalyst must be provided in order to give a reference (acetaldehyde is quite reactive and the typical sampling system is solid phase chemisorption with 2,4-dinitrophenylhydrazine and HPLC analysis).

Rows 372-375 - It is somewhat misleading to directly relate the acetaldehyde and NOx removal because they are obtained with very different systems (acetaldehyde with a sealed reactor and NOx with a continuous flow laminar reactor) and thus the measured degradation has a very different meaning. Moreover, the data demonstrate also a net production of NO2in all NO degradation experiments that should be noted and commented in the text.

[1]       M. Gallus et al. “Photocatalytic abatement results from a model street canyon”, Environ Sci Pollut R,22 (2015) 18185-18196.

Round 2

Reviewer 1 Report

The rebuttal made by the authors is not convincing, because advancement in catalysis research field is unclear. Thus, this manuscript is more suitable for the journal treating water treatment than Catalysts. 

Reviewer 2 Report

This is  a valuable first manuscript from the Korea research group towards the development of  a photoactive anatase TiO2 from algae bloomed surface water.  The changes introduced in the revised version are valuable indeed. Thus, the manuscript is fine as it is now. The authors are however encouraged,to proceed with a more quantum efficiency study in a future contribution.